# Homocysteine and Glaucoma

**DOI:** 10.3390/ijms241310790

**Published:** 2023-06-28

**Authors:** Joshua Washington, Robert Ritch, Yutao Liu

**Affiliations:** 1Department of Cellular Biology and Anatomy, Medical College of Georgia, Augusta University, Augusta, GA 30912, USA; jowashington@augusta.edu; 2New York Eye & Ear Infirmary, New York, NY 10003, USA; ritchmd@glaucoma.net; 3James & Jean Culver Vision Discovery Institute, 4 Center for Biotechnology and Genomic Medicine, Medical College of Georgia, Augusta University, Augusta, GA 30912, USA

**Keywords:** homocysteine, glaucoma, aqueous humor, primary open-angle glaucoma, exfoliation glaucoma

## Abstract

Elevated levels of homocysteine (Hcy), a non-proteinogenic amino acid, may lead to a host of manifestations across the biological systems, particularly the nervous system. Defects in Hcy metabolism have been associated with many neurodegenerative diseases including glaucoma, i.e., the leading cause of blindness. However, the pathophysiology of elevated Hcy and its eligibility as a risk factor for glaucoma remain unclear. We aimed to provide a comprehensive review of the relationship between elevated Hcy levels and glaucoma. Through a systemic search of the PubMed and Google Scholar databases, we found that elevated Hcy might play an important role in the pathogenesis of glaucoma. Further research will be necessary to help clarify the specific contribution of elevated Hcy in the pathogenesis of glaucoma. A discovery and conceptual understanding of Hcy-associated glaucoma could be the keys to providing better therapeutic treatment, if not prophylactic treatment, for this disease.

## 1. Introduction

Homocysteine (Hcy) is a non-proteinogenic amino acid discovered by Butz and du Vigneaud in 1932 [1]. Hcy was originally considered to be a metabolite of methionine. Extensive research has demonstrated that Hcy could be involved in processing DNA, RNA, and other biochemical material [2,3]. Hcy is involved in these processes via two biochemical pathways: the methionine cycle and the transsulfuration sequence [4,5]. Genetic anomalies, pharmacotherapy, malnutrition, and substance abuse may compromise the methionine cycle or transsulfuration sequence, leading to pathologic, intracellular levels of Hcy [3,6,7]. Because Hcy is found in every cell type, the upregulation of Hcy can lead to a host of manifestations, namely in the cardiovascular and nervous systems [8,9,10,11,12,13].

Hcy is naturally found in the blood with a normal level of 5–15 μmol/L in circulation. Elevated levels of Hcy (>15 μmol/L) could be identified in many individuals, including mild, intermediate, and severe levels. These elevated levels of homocysteine are termed hyperhomocysteinemia (HHcy). Mild levels of HHcy in the blood are defined as 15–30 μmol/L of Hcy, intermediate levels as 30–100 μmol/L of Hcy, and severe levels as over 100 μmol/L of Hcy [14,15,16,17,18]. Hcy levels in body fluids (for example, serum, plasma, tears, and aqueous humor) and cells could be measured using enzyme-linked immunosorbent assays (ELISA), fluorescence polarization immunoassays (FPIA), competitive chemiluminescent enzyme immunoassays (CCEI), AU 500 automatic biochemical analyzer (AU), and mass spectrometry.

HHcy is a risk factor in the pathophysiology of atherosclerosis, vascular fibrosis, epilepsy, and intellectual disability [9,10,19,20,21,22]. HHcy has even been confirmed as a risk factor for ocular pathologies such as age-related macular degeneration (AMD) and anophthalmia [23,24]. However, whether HHcy serves as a risk factor for other oculopathies remains inconclusive, notably for glaucoma. Glaucoma is the leading cause of irreversible blindness worldwide, so an advanced understanding of HHcy in glaucoma can lead to improved therapeutic treatments and the introduction of prophylactic ones as well [25]. Using both human and animal studies, we aimed to provide a comprehensive review on the relationship between HHcy and glaucoma to ultimately answer the big question: is HHcy a risk factor for glaucoma?

## 2. Homocysteine

In 1932, Butz and du Vigneaud exposed methionine, an essential amino acid, to different concentrations of sulfuric acid [1]. At the optimal concentration, methionine yielded 2-amino-4-mercaptobutyric acid, now known as homocysteine. Since then, Hcy has been considered a non-proteinogenic amino acid that accepts methyl groups from N-5-methyltetrahydrofolate (methylTHF), betaine, or choline in order to form methionine [4,5]. Hcy metabolism in vivo involves two pathways: the methionine cycle and the transsulfuration sequence, which are responsible for regenerating methionine and catabolizing surplus Hcy, respectively (Figure 1) [26]. These pathways largely depend on S-adenosylmethionine (SAM), a universal methyl donor to metabolites such as DNA, RNA, and proteins in many biochemical pathways [2,3]. Once these biochemical pathways are sufficiently supplied with methyl groups, the activity of the methionine cycle declines and, along with it, Hcy levels. As a result, SAM initiates Hcy catabolism by activating cystathionine β-synthase (CBS) [4]. With the help of its coenzyme pyridoxal phosphate (PLP), CBS uses Hcy as a substrate to form cystathionine [27]. Specifically, CBS replaces the beta-hydroxyl group of serine with Hcy via its thiol group in order to form a cystathionine that eventually gets excreted into the urine as inorganic sulfate [4,27,28].

## 3. Hyperhomocysteinemia

Sulfur-containing amino acids such as Hcy are prone to generate reactive oxygen species (ROS), leading to oxidative stress within the cell [29,30]. Specifically, Hcy upregulates NADPH oxidase which, in turn, catalyzes the formation of superoxide, a specific ROS. As a preventive measure, the transsulfuration sequence catabolizes surplus Hcy [4]. When this surplus of Hcy goes unregulated, it leads to HHcy in vivo. Though the mechanism is not completely understood, work suggests that Hcy enters blood circulation through multiple cysteine transporters [31].

Depending on which is involved, defects in CBS, methylenetetrahydrofolate reductase (MTHFR), methylation enzymes, and intracellular cobalamin will lead to various levels of severity in HHcy [4]. The severe form of HHcy stems from defects in CBS or MTHFR. Defective CBS stems from genetic defects while defective MTHFR stems from folate deficiencies and/or genetic defects. Mechanistically, CBS-induced HHcy drives the methionine cycle forward (Figure 1), including the upregulation of SAM which, in excess, inhibits MTHFR, an enzyme catalyzing the formation of methylTHF. Since methylTHF is needed to transform Hcy into methionine, defective MTHFR can also lead to HHcy. Furthermore, both CBS- and MTHFR-induced HHcy are exacerbated when considering the transsulfuration sequence. When SAM is in short supply, the transsulfuration sequence cannot be activated due to the scarcity of methionine: the SAM precursor. Regarding HHcy with an intermediate severity, methylation enzymes could be defective, namely methionine adenosyltransferase (MAT) I/III, glycine N-methyltransferase (GNMT1), S-adenosyl hydrolase (SAHH), and adenosine kinase (ADK) [18]. 

HHcy with a mild severity stems from a cobalamin deficiency or from a defect in any enzyme responsible for catalyzing cobalamin (inactive vitamin B_12_) into its active form: methylcobalamin (Figure 2) [4,32]. HHcy stemming from reduced cobalamin levels exhibits a mild severity because of the relative bioavailability of SAM. According to the methyl trap hypothesis, SAM catabolism is inhibited by the accumulation of methylTHF, thus counteracting SAM depletion that stemmed from reduced cobalamin levels (Figure 3) [33,34]. As a result, Hcy levels decrease while transsfuluration activity increases, thereby justifying the mild severity of HHcy associated with reduced cobalamin levels.

In addition to genetic anomalies, HHcy may arise from malnutrition and renal failure [3,6]. Because methionine is an essential amino acid, a diet that lacks methionine may lead to HHcy. Methionine is abundant in animal proteins, so excessively consuming meat, eggs, and milk can upregulate the methionine cycle in which Hcy is involved. Interestingly, an insufficient amount of animal proteins could also lead to HHcy. Aside from methionine, animal protein also contains vitamin B_12_ and folate, which are important coenzymes involved in Hcy metabolism [35]. So, their deficiencies underscore the inversely proportional relationship between levels of vitamin B_12_/folate and Hcy. Furthermore, nicotine and alcohol will, respectively, compromise folate and cobalamin production which translates into the upregulation of Hcy. Drugs such as cholestyramine and metformin are associated with vitamin and folate malabsorption in the gastrointestinal tract. Regarding renal clearance, a defect in Hcy excretion may lead to HHcy, although the exact mechanism is not understood [6].

## 4. Manifestations of Hyperhomocysteinemia

HHcy is associated with many pathologies of the endocrine, renal, reproductive, and gastrointestinal systems, including homocystinuria, chronic renal failure (CRF), hypothyroidism, insulin-resistant diabetes, polycystic ovarian syndrome (PCOS), gastrointestinal disorders, and pregnancy complications [36,37,38,39,40,41]. However, the most prominent effects of HHcy lay within the cardiovascular and nervous systems.

### 4.1. Cardiovascular Associations with Hyperhomocysteinemia

The proximal consequence of HHcy is oxidative stress which, in turn, leads to a host of cardiovascular diseases including but not limited to atherosclerosis, thrombosis, vasoconstriction, and vascular fibrosis. Atherosclerosis results from the upregulation of adhesion molecules and HMG CoA reductase [22]. Thrombosis and vasoconstriction result from the upregulation of coagulants such as factor V and the downregulation of anticoagulants such as thrombomodulin [42,43,44]. Vascular fibrosis results from the activation of matrix metalloproteinases (MMPs) that disrupt the extracellular matrix (ECM) [21]. Lastly, compromises in methylTHF and pyridoxal phosphate levels result in hypertension and coronary artery disease, respectively [45,46]. Understandably, endothelial cell dysfunction is thought to be the basis for HHcy-induced cardiovascular disease [19].

### 4.2. Neurodegenerative Associations with Hyperhomocysteinemia

In addition to cardiovascular pathology, elevated Hcy levels were associated with neurodegenerative manifestations such as dementia, epilepsy, and retardation. In Alzheimer’s disease (AD), HHcy causes the formation of amyloid plaques by inactivating the enzyme responsible for their degradation: secretase [20]. Epilepsy is caused by the overactivation of the N-methyl-D-aspartate (NMDA) receptor. Because homocysteine is an NMDA receptor agonist, elevated Hcy may induce epilepsy via an excessive calcium influx, leading to defective neuronal signaling [9,10]. Regarding mental retardation, data show that it stems from the severe form of HHcy induced by defects in CBS [8]. Besides these causal links, HHcy has been correlated with autism, stroke, multiple sclerosis (MS), depression, cerebral seizures, myelopathy, psychosis, and Parkinson’s disease (PD) [47,48,49].

Furthermore, upregulated Hcy is associated with ocular pathologies, including but not limited to keratoconus, Behcet’s disease, retinal detachment, retinoblastoma, and central retinal vein occlusion (CRVO) [50,51,52,53,54]. Elevated Hcy levels are associated with more common oculopathies such as peripheral diabetic retinopathy (PDR), AMD, cataracts, and myopia [47,55,56,57,58,59,60]. PDR, a complication of diabetes, has a host of factors associated with it: renal dysfunction, elevated A1C levels, increased systolic blood pressure, and the duration of diabetes [61]. However, these risk factors either act independently of the Hcy–PDR relationship or cannot explain it at all, thus furthering the possibility of Hcy being a novel biomarker for PDR. Studies on myopia and cataracts have suggested a potential role of elevated Hcy in their pathogenesis [57,62]. Myopia is a state of refraction in which parallel light rays are focused in front of the retina instead of onto the retina itself [63]. Myopia can either be physiologic or pathologic, but in this context, myopia is referenced pathologically. Pathologically, myopia results from the abnormal lengthening of the eye and is also associated with scleral thinning [64]. Elevated Hcy levels have been connected to this pathogenesis after discovering that myopia was prevented in patients with homocystinuria when they were treated with Hcy-lowering therapy early [57]. Cataracts are also associated with upregulated Hcy, as evidenced by their presence in cataract patients [55,62,65]. With the help of avian and rodent animal models, the association of oculopathies such as AMD and anophthalmia with upregulated Hcy has been shown to be causal [23,24]. The associations in these animal models are more than mere due to the overlapping similarities between human and rodent eye structures. Conclusively, an association between homocysteine and human oculopathies seems even more plausible [66].

Despite the vast research on links between HHcy and various ocular pathologies, of particular importance is the relationship between HHcy and glaucoma: an ocular neuropathy that serves as the leading cause of irreversible blindness worldwide [25]. To appreciate such an impact, a foundation of eye structure is vital.

## 5. Eye Structure

The eye is organized into an outer, middle, and inner layer [67]. The outer layer consists of the sclera and the cornea which help to prevent infection and mechanical trauma (Figure 4). The cornea also refracts light that traverses the central, open region of the iris and refracts onto the lens. Refraction of light onto the lens and the retina ultimately occurs via contractions and relaxations of the iris, ciliary body, and choroid, i.e., the constituents of the middle layer. Within the ciliary body specifically, aqueous humor is produced from ocular perfusion pressure (OPP) [68]. It drains out via two systems: trabecular meshwork and uveoscleral outflow (Figure 5). The inner layer is made up of the retinal pigment epithelium (RPE), photoreceptors, and other neuronal subtypes: bipolar cells, horizontal cells, amacrine cells, Müllerian glial cells, and retinal ganglion cells (RGCs) (Figure 6). The RPE maintains the health of photoreceptors while factors such as brain-derived neurotrophic factor (BDNF) maintain the health of the other neuronal subtypes [66].

## 6. Glaucoma

Glaucoma is a group of optic neuropathies that share a similar feature: the degeneration of RGCs [66]. As a result, the optic disc disappears which clinically manifests as vision loss. Glaucoma may occur spontaneously without an association to a known cause (primary glaucoma) or it can be a sequela to an underlying disease (secondary glaucoma). Primary glaucoma consists of open-angle glaucoma (POAG), angle-closure glaucoma (PACG), and congenital glaucoma (PCG). POAG can be further categorized into high tension glaucoma (HTG) and normal tension glaucoma (NTG). HTG is quantified as an intraocular pressure (IOP) > 21 mmHg while that of NTG is < 21 mmHg. Excluding NTG, all glaucomatous neuropathies are associated with increased IOP [66,69,70,71]. These neuropathies include all forms of secondary glaucoma: neovascular glaucoma, pigmentary glaucoma, exfoliation glaucoma, and uveitic glaucoma [72,73,74,75]. The mechanisms behind these neuropathies are incompletely understood. POAG is differentiated from the other neuropathies by the sole presence of an open angle of the anterior chamber coupled with ocular hypertension in HTG patients. For NTG, diagnostic factors are similar, with ocular hypertension being the exception. As the name suggests, NTG has a normal IOP without ocular hypertension. In PACG, the angle of the anterior chamber has an abnormally closed appearance coupled with ocular hypertension. In PCG, pathogenesis is tied to a developmental defect of the anterior segment itself. Particularly, an imperforate film is located on the surface of the trabecular meshwork which is thought to prevent the outflow of aqueous humor and ultimately cause ocular hypertension [76]. Pathological contact with the trabecular meshwork leads to ocular hypertension: the primary risk factor for all types of glaucoma [66].

Worldwide, 70 million people have glaucoma, and this number is projected to be 111.8 million by 2040 [77,78]. When considering the most common form of glaucoma, this review focuses on POAG so that slowing, plateauing, and reversing the incidence rate of glaucomatous disease can be done more efficiently. 

## 7. Risk Factors for Glaucoma

Alongside human epidemiology, various animal models have identified several risk factors for glaucoma, including advanced aging, myopia, a family history of glaucoma, African or Hispanic ancestry, and elevated IOP. IOP is the only modifiable risk factor in a clinic. In response to elevated IOP, the pressure gradient across the lamina cribrosa increases and compresses the RGCs. This may impair the axonal transport of trophic factors which ultimately leads to neuronal death by trophic insufficiency [66]. Other risk factors for glaucoma include OPP, primary vascular dysregulation (PVD), and systemic hypotension. Interestingly, they are complexly linked to each other and to IOP [79,80]. OPP can be caused by a decrease in systemic hypotension, the presence of PVD, and an increase in IOP [79]. Though physiological, systemic hypotension can also occur pathologically during the night [81,82]. Since OPP and mean arterial pressure (MAP) are proportional to each other, nocturnal hypotension can cause a decrease in OPP. Unlike MAP, OPP is inversely proportional to IOP: a factor associated with PVD. PVD is the idiopathic inability to adapt to changes in blood flow. In PVD, IOP increases when stiff vessels cannot respond to a change in blood flow [83]. This can ultimately cause ischemia-reperfusion damage of the optic nerve via free radicals [84]. Conclusively, systemic hypotension, PVD, and IOP are interconnected with respect to their effects on OPP.

Genetics and senescence are also recognized as risk factors in glaucoma [85,86]. An elevated IOP at baseline and a thinner cornea have both been tied to heritability. At the molecular level, more than 127 genes/loci have been associated with glaucoma while those of elevated IOPs at baseline were associated with more than 150 [86,87,88,89,90]. The pathogenesis of central corneal thickness (CCT)-induced glaucoma still needs to be fully understood. However, work suggests that a corneal layer is displaced by IOP fluctuations which, in turn, leads to the loss of adjacent axons and damage to the lamina cribrosa [91,92]. Regarding senescence, researchers report that after 60 years of age, the risk of developing glaucoma becomes significant and continues to increase with each decade of life [93]. Conclusively, age and a family history of glaucoma are considered risk factors for the disease.

Of note are other contributing factors of POAG: ischemia, hypoxia, excessive stimulation of NMDA, oxidative stress, free radicals, and aberrant immunity [66]. Furthermore, the progression of glaucoma occurring iatrogenically is worthy of mention because antihypertensive treatments and ocular surgery are also risk factors for the progression of glaucoma [94,95,96,97].

## 8. The Relationship between Homocysteine and Glaucoma

Studies have associated HHcy with different types of glaucoma [98,99,100,101,102,103,104]. Using *homocysteine* and *glaucoma* as key words, we searched PubMed in order to summarize different studies of HHcy with different types of glaucoma in Table 1, Table 2, Table 3 and Table 4.

Roedl et al. found that 39 POAG patients had higher Hcy levels in their aqueous humor and plasma than 39 controls [104]. Micheal et al. found that 122 PACG patients had higher Hcy levels in their serum and plasma than 143 controls [102]. Bleich et al. found that 29 exfoliation glaucoma patients had higher Hcy levels in their aqueous humor and plasma compared to 31 controls [98]. Cumurcu et al. used CCEI and found that 24 exfoliation glaucoma patients had higher Hcy levels in their serum compared to 19 controls [101]. Yücel et al. applied FPIA assays and found that 40 neovascular glaucoma patients had higher Hcy levels in their plasma compared to 30 controls [100]. These positive associations of HHcy with glaucoma are substantiated when considering retinal fiber nerve layer (RNFL) defects are also associated with Hcy [115,116]. In contrast, some studies identified no association between Hcy levels and glaucoma. For example, Wang et al. found that 55 Chinese POAG patients showed similar Hcy levels in their plasma compared to 39 controls [109]. Cumurcu et al. showed that 25 POAG and 18 NTG patients had similar Hcy levels in their serum compared to the 19 controls [101]. Rossler et al. indicated that 42 NTG patients showed similar Hcy levels in their plasma compared to 42 controls [114]. 

The disagreement among these reports could stem from differences in laboratory techniques (i.e., HPLC, ELISA, etc.), genetic factors in different ethnic populations, study-specific criteria for surgical eligibility, sample sizes, and sample media (i.e., aqueous humor, plasma, serum) [98,100,101,102,103,104,105,106,107,108,109,110,111,112,113,114,115,116,117,118,119]. Measuring Hcy using different lab techniques, sample media, and other factors could cause results to be incomparable. As we mentioned earlier in the review, environmental factors such as nutritional status and diet could also lead to HHcy in selected patients or controls in any of these studies as well [3]. Conclusively, the association of homocysteine with glaucoma is multifactorial. 

The potential relationship of HHcy with congenital, pigmentary, and uveitic glaucoma is also of note. Hcy levels in these glaucoma types have not been studied as extensively as the aforementioned ones, thereby emphasizing the need for further investigation into Hcy levels within these less common types of glaucoma.

Similar to the individual case-control studies, meta-analyses also reported mixed results on the relationship between HHcy and glaucoma. From their respective systemic searches, Ajith and Ranimenon reported that some studies found a proportional relationship between intravitreal Hcy levels and proliferative retinopathy diagnoses, while others found no such association [118]. Ritch reported an association of HHcy with PXG exists while Zacharaki et al. found the opposite [119,120]. It remains unclear whether the genetic mutations or variations in the MTHFR gene could be causal to HHcy associated with risks of glaucoma [121,122,123,124]. These disagreements further highlight the complexity of a possible Hcy–glaucoma association.

## 9. The Pathophysiology of Homocysteine in Primary Open Angle Glaucoma

It is known that increased intracellular levels of Hcy can increase arterial pressure [46]. Additionally, increases in arterial pressure contribute to the defect in OPP which could lead to increased IOP and, ultimately, POAG. Due to the interconnectedness of Hcy, arterial pressure, OPP, and IOP, associating arterial pressure with Hcy-associated POAG seems plausible. However, evidence to the contrary is strongly supported as arterial pressure was demonstrated to have no significant effect on IOP [68]. Because an optimal IOP is mainly achieved through resistance mechanisms in aqueous drainage systems, aqueous inflow seems to be a negligible contributing factor. This explains the lack of correlations between HHcy and POAG as well as the ineffectiveness of antihypertensive therapy on patients with POAG [109,125,126]. Undoubtedly, progress in understanding the mechanism for HHcy-associated POAG seems stagnant.

Nevertheless, other avenues to discovery can be made when focusing directly on the retina intrinsically. If epithelial dysfunction of the RPE occurs, RGCs could eventually die due to poor stimulation or support. Specifically, photoreceptors lose viability in the absence of the RPE. Because photoreceptors initiate visual transduction, their absence could result in the death of RGCs due to the lack of stimulation earlier in the pathway. Moreover, because homocysteine metabolism occurs in every cell, its defective metabolism within both the RGCs and optic nerve can be another possibility behind the mechanism of HHcy-associated POAG [127,128]. In fact, a functional analysis of the *mthfr* mutant mice demonstrated that HHcy in the retina of mice led to increased RGC death [129]. By definition, glaucoma is the term for RGC death which signifies that MTHFR is a genetic risk factor for POAG [129]. Altogether, genetic associations between glaucoma and variants of MTHFR sequences have been extensively investigated to suggest their associations with POAG [130,131,132,133].

In discovering the pathogenesis of HHcy-associated glaucoma, further research on its mechanism involved several factors: nicotinamide (NAM), NRF2, the endoplasmic reticulum (ER), mitochondria, and the N-methyl-D-aspartate (NMDA) receptor. In substantiating NAM-mediated neuroprotection against glaucoma, Tribble et al. found that the retinal metabolome was altered in subjects who only had elevated IOP for three days: a time point at which IOP is high but RGC loss is not detectable [134,135,136]. Because homocysteine metabolism was included in that altered retinal metabolome, it supports the notion that HHcy plays a causal role in the pathogenesis of glaucoma [135]. Adding to that notion was the essential component of an NRF2 discovery: intravitreal Hcy. To help discover the inversely proportional relationship between NRF2 levels and RGC thinness, Navneet et al. investigated the effects of acute Hcy exposure on retinal structures [137]. The results showed that direct intravitreal injection of Hcy in *nrf2*-null mice led to the decreased viability of their RGCs [137]. This, in combination with Hcy upregulating an NRF2-antioxidant pathway, suggests NRF2 is a component of Hcy-associated glaucoma [136]. Videlicet, NRF2 seems to play a cytoprotective response for conditions in which HHcy is present. Furthermore, in assessing the behavior and biochemical activity of CBS-deficient mice, upregulation of the protein kinase r-like ER kinase (PERK) pathway as well as Opa1 and Fis1 suggests that the ER and mitochondria are also components of Hcy-associated glaucoma [135,136]. Lastly, because Ganapathy et al. suggested that Hcy-mediated stimulation of NMDA receptors leads to RGC death, overstimulation of these NMDA receptors could also be considered when understanding the pathogenesis of Hcy-associated glaucoma [137]. Conclusively, discovering the pathogenesis of Hcy-associated glaucoma seems more of a possibility when considering the contributions of the aforementioned factors. In doing so, suggesting that Hcy can serve as the causative agent rather than a mere biomarker of Hcy-associated glaucoma seems plausible.

## 10. Conclusions

The association of increased Hcy levels with glaucoma risk has been suggested by population studies, animal model-based investigations, and cellular/molecular experiments. However, knowledge of the underlying cellular and molecular pathways related to the pathogenesis of glaucoma remains unclear. This promulgates the need to continue research on HHcy and glaucoma in order to identify novel molecular targets that treat and prevent glaucoma. More than 70 million people are affected by glaucoma worldwide, and 3 million of them are American [77]. HHcy may interact with glaucoma-relevant genetic and environmental factors contributing to the development of glaucoma. It might be beneficial to identify those individuals with HHcy and to reduce their systemic Hcy levels via diet, nutrition supplement, or medications, followed by annual ophthalmic exams for glaucoma prevention. Glaucoma patients with HHcy could seek additional professional help to reduce their Hcy levels while under the standard care by their glaucoma specialists. The treatment of HHcy in these glaucoma patients could provide potential novel approaches for clinical treatment. An advanced understanding of the mechanism behind HHcy-associated POAG is of utter importance because it will help slow the progression of the disease, if not prevent its inception.

## Figures and Tables

**Figure 1 ijms-24-10790-f001:**
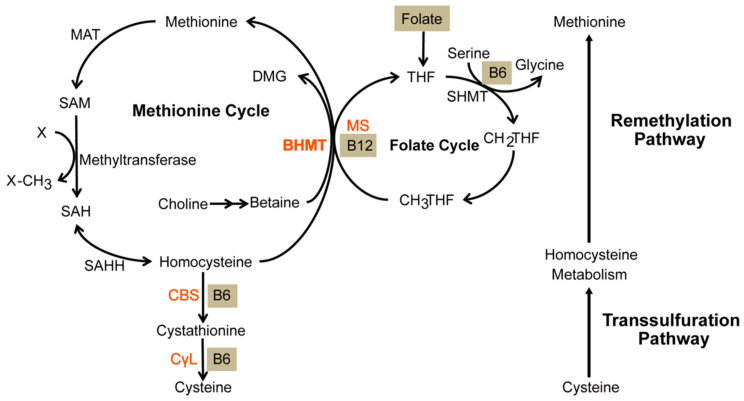
Anabolism of methionine and catabolism of homocysteine. The remethylation pathway consists of both the methionine and folate cycle. It exists to anabolize methionine. During this process, a methyl group is transferred from homocysteine. Subsequently, it enters the transsulfuration pathway whereby cystathionine β-synthase (CBS) catabolizes it into cystathionine that will eventually be catabolized into a waste product. Homocysteine is metabolized to methionine by remethylation and cystathionine by transsulfuration. Gray lettering represents coenzymes. BHMT—betaine-homocysteine S-methyltransferase, DMG—dimethylglycine, MAT—methionine adenosyltransferase, SAM—S-adenosylmethionine, SAH—S-adenosylhomocysteine, SAHH—S-adenosylhomocysteine hydrolase, MS—methionine synthase, THF—tetrahydrofolate, SHMT—serine hydroxymethyltransferase, CH_2_THF—methylene tetrahydrofolate, CH_3_THF—methyl tetrahydrofolate, CBS—cystathionine β-synthase, CγL—cystathionine γ-lyase. Figure 1 is sourced from [26].

**Figure 2 ijms-24-10790-f002:**
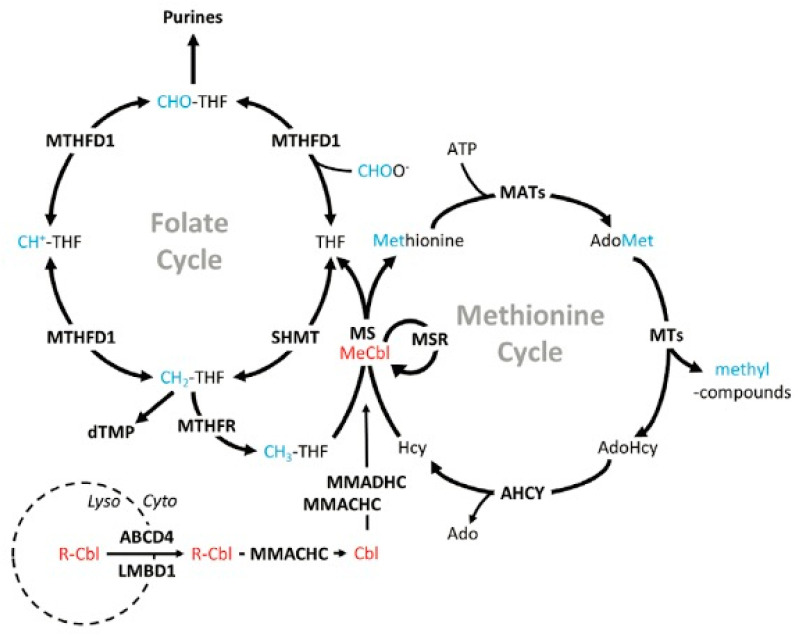
Cobalamin synthesis. Red labels indicate enzymes in the cobalamin-synthesis pathway. Cobalamin is transported out of the lysosome and, through a series of reactions in the remethylation pathway, it is modified into methylcobalamin (active vitamin B_12_). Blue lettering represents methyl groups while bold lettering represents proteins. ABCD4—ATP-binding cassette subfamily D member 4, Ado—adenosine, AdoHcy—adenosylhomocysteine, AdoMet—S-adenosylmethionine, AHCY—adenosylhomocysteinase, ATP—adenosine triphosphate, Cbl—cobalamin (no upper axial ligand attached), CH^+^-THF—5,10-methenyltetrahydrofolate, CH_2_-THF—5,10-methylenetetrahydrofolate, CH_3_-THF—5-methyltetrahydrofolate, CHO-THF—10-formyltetrahydrofolate, CHOO^−^—formate, dTMP—deoxythymidine monophosphate, Cyto—cytosol, LMBD1—lipocalin-1-interacting membrane receptor domain-containing 1, Lyso—lysosome, MATs—methionine adenosyltransferase(s), MeCbl—methylcobalamin, MMACHC—methylmalonic aciduria cblC type (with homocystinuria), MMADHC—methylmalonic aciduria cblD type (with homocystinuria), MS—methionine synthase, MSR—methionine synthase reductase, MTHFD1—methylenetetrahydrofolate dehydrogenase 1, MTHFR—dmethylenetetrahydrofolate reductase, MTs—methyltransferase(s), R-Cbl—upper axial ligand (eg, cyano-, hydroxo-) attached to cobalamin, SHMT—serine hydroxymethyltransferase. Figure 2 is sourced from [32].

**Figure 3 ijms-24-10790-f003:**
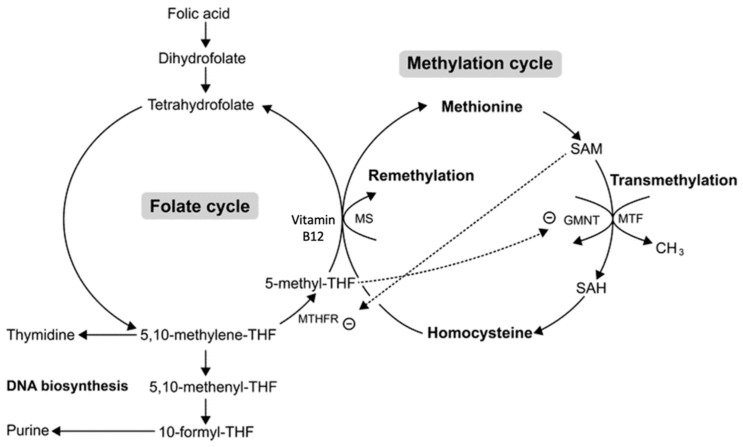
Methylcobalamin-induced mild hyperhomocysteinemia (HHcy). A methylcobalamin (Vitamin B12) deficiency leads to HHcy from the methylation cycle and a methylTHF surplus from the folate cycle. This surplus of methyl THF leads to the accumulation of SAM due to the inhibition of glycine N-methyltransferase (GMNT). Although HHcy can stem from a deficiency in methylcobalamin, the excess methylTHF formed from that deficiency counteracts HHcy via SAM accumulation, thereby causing HHcy to be mild. THF—tetrahydrofolate, SAM—S-adenosyl methionine, SAH—S-adenosyl homocysteine, MS—methionine synthase, MTF—methyltransferase, MTHFR—methylene-THF reductase. Figure 3 is sourced from [34].

**Figure 4 ijms-24-10790-f004:**
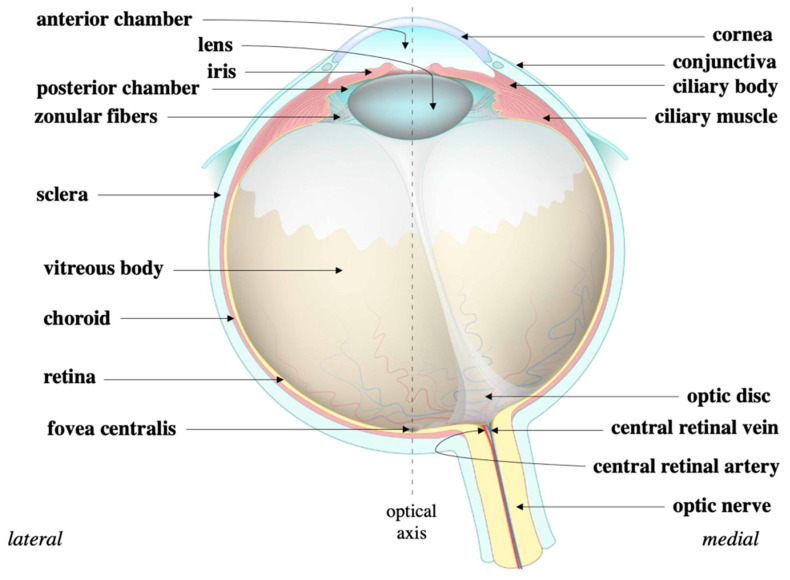
Eye structure. The cornea is continuous with the sclera that circumvents the eye. The sclera protects the eye while the cornea refracts light onto the lens. During that process, light traverses the iris and eventually the ciliary body to ultimately refract onto the retina. Source: Anatomy of the eye, courtesy of Amboss.

**Figure 5 ijms-24-10790-f005:**
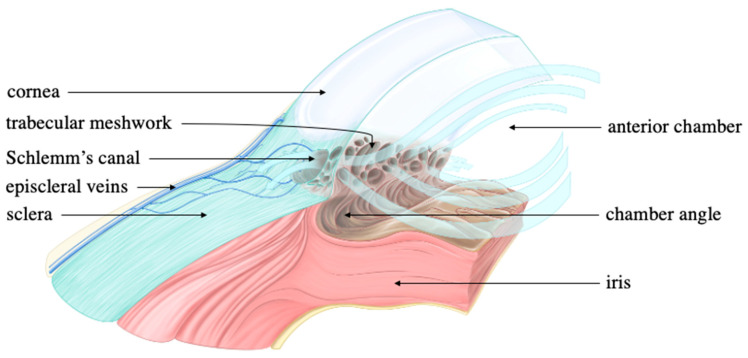
Trabecular and uveoscleral outflow. Thickened light blue arrows represent aqueous flow while thinned black arrows are associated with ocular structures. The thickened light blue arrows extend from the anterior chamber and into the trabecular meshwork. This directionality marks the conventional (trabecular) outflow of the aqueous humor. Though arrows are not drawn to represent the unconventional (uveoscleral) outflow, aqueous humor can also exit from the chamber angle and directly into the sclera to ultimately enter the episcleral veins. Source: Anterior chamber angle, courtesy of Amboss.

**Figure 6 ijms-24-10790-f006:**
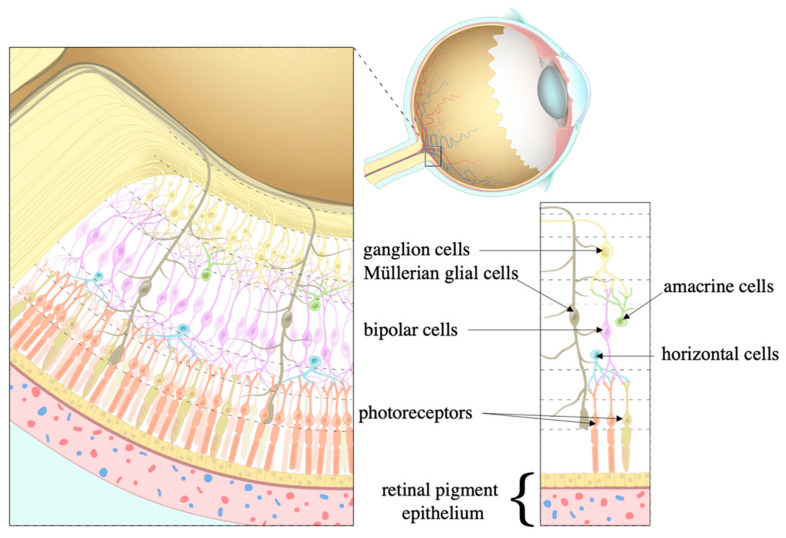
Cell types within the retina. Once light hits the retina, it is processed by the photoreceptors that pass information to the ganglion cells to which bipolar, horizontal, amacrine, and Müllerian glial cells contribute. The retinal pigment epithelium (RPE) maintains this processing. *Source: Layers of the retina, courtesy of Amboss*.

**Table 1 ijms-24-10790-t001:** The association of homocysteine levels with POAG.

Studies	Glaucoma Type	Location (Race)	Controls *^1^ (Gender, Age (a)) Test Group (Gender, Age (a))	Hcy Level in Controls	Hcy Level in Test Group	Significance (*p* Value)	Sample Types	Lab Technique	Surgical Eligibility
[105]	POAG	Nanchang, China (N/A)	53 controls (30 m + 23 f, a-62) 41 in test group (22 m + 19 f, a-59)	10.82 μmol/L	14.44 μmol/L	*p* < 0.01	P	AU	Inclusions: N/A Exclusions: N/A
[105]	POAG	Nanchang, China (N/A)	53 controls (30 m + 23 f, a-62) 41 in test group (22 m + 19 f, a-59)	0.69 μmol/L	1.60 μmol/L	*p* < 0.01	AH	AU	Inclusions: N/A Exclusions: N/A
[106]	POAG	Alicante, Spanish (N/A)	75 controls (17 m + 58 f, a-44) 48 in test group (23 m + 25 f, a-50)	2.6 μmol/L	7.6 μmol/L	*p* = 0.002	P	CCEI	Inclusions: N/A Exclusions: N/A
[107]	POAG	Rome, Italy (N/A)	40 controls (21 m +19 f, a-69) 40 in test group (22 m + 18 f, a-69)	13.12 μmol/L	13.91 μmol/L	*p* = 0.56	P	CCEI	Inclusions: N/A Exclusions: intraocular surgery within 12 months, laser surgery within 3 months
[108]	POAG	Sydney, Australia (Caucasian)	42 controls (16 m + 26 f, a-70) 39 in test group (17 m + 22 f, a-72)	9.82 μmol/L	11.21 μmol/L	*p* < 0.01	P	FPIA	Inclusions: laser trabeculoplasty, glaucoma filtration surgery Exclusions: N/A
[104]	POAG	Nuremberg, Germany (Caucasian)	39 controls (18 m + 21 f, a-71) 39 in test group (17 m + 18 f, a-69)	1.12 μmol/L	1.76 μmol/L	*p* < 0.001	AH	HPLC	Inclusions: N/A Exclusions: prior ocular surgery
[104]	POAG	Nuremberg, Germany (Caucasian)	39 controls (18 m + 21 f, a-71) 39 in test group (17 m + 18 f, a-69)	10.46 μmol/L	13.93 μmol/L	*p* < 0.001	P	HPLC	Inclusions: N/A Exclusions: prior ocular surgery
[104]	POAG	Erlangen-Nuremberg, Germany (Caucasian)	39 controls (18 m + 21 f, a-71) 39 in test group (17 m + 18 f, a-69)	1.12 μmol/L	1.76 μmol/L	*p* < 0.001	AH	HPLC	Inclusions: N/A Exclusions: prior ocular surgery
[101]	POAG	Tokat, Turkey (N/A)	19 controls (5 m + 14 f, a-57) 25 in test group (7 m + 18 f, a-56)	8.40 μmol/L	9.22 μmol/L	*p* > 0.05	S	CCEI	Inclusions: undergoing ocular surgery Exclusions: N/A
[102]	POAG	Pakistan (Punjabis/Pathans)	143 controls (73 m + 70 f, a-49) 122 in test group (88 m + 34 f, a-50)	10.00 μmol/L	20.48 μmol/L	*p* < 0.05	S	ELISA	Inclusions: N/A Exclusions: N/A
[109]	POAG	California, United States (N/A)	39 controls (5 m + 34 f, a-73) 55 in test group (33 m + 22 f, a-75)	14.81 μmol/L	14.90 μmol/L	*p* = 0.93	P	HPLC	Inclusions: undergoing ocular surgery Exclusions: N/A

Note: AH—aqueous humor; ELISA—enzyme-linked immunosorbent assay; P—plasma; FPIA—fluorescence polarization immunoassay; S—serum; CCEI—competitive chemiluminescent enzyme immunoassay; TF—tear fluid; m—male; f—female; a—average of ages; AU—AU 500 automatic biochemical analyzer; N/A—not available from the original paper. ^*1^—controls are subjects without POAG.

**Table 2 ijms-24-10790-t002:** The association of homocysteine levels with PEXG.

Studies	Glaucoma Type	Location (Race)	Controls *^1^ (Gender, Age (a)) Test Group (Gender, Age (a))	Hcy Level in Controls	Hcy Level in Test Group	Significance (*p* Value)	Sample Types	Lab Technique	Surgical Eligibility
[110]	PEXG	Batman, Turkey (N/A)	35 controls (18 m + 17 f, a-67) 24 in test group (10 m + 14 f, a-67)	9.9 μmol/L	15.4 μmol/L	*p* < 0.001	P	HPLC	Inclusions: N/A Exclusions: N/A
[107]	PEXG	Rome, Italy (N/A)	40 controls (21 m + 19 f, a-69) 36 in test group (25 m +11 f, a-70)	13.12 μmol/L	16.55 μmol/L	*p* < 0.0007	P	CCEI	Inclusions: N/A Exclusions: intraocular surgery within 12 months, laser surgery within 3 months
[108]	PEXG	Sydney, Australia (Caucasian)	42 controls (16 m +26 f, a-70) 48 in test group (17 m + 31 f, a-74)	9.82 μmol/L	11.77 μmol/L	*p* < 0.05	P	FPIA	Inclusions: laser trabeculoplasty, glaucoma filtration surgery Exclusions: N/A
[111]	PEXG	Nuremberg, Germany (N/A)	70 controls. (33 m +37 f, a-68) 70 in test group (32 m + 38 f, a-70)	10.45 μmol/L	13.77 μmol/L	*p* < 0.001	P	HPLC	Inclusions: undergoing glaucoma or cataract surgery Exclusions: N/A
[101]	PEXG	Tokat, Turkey (N/A)	19 controls (5 m + 14 f, a-57) 24 in test group (10 m + 14 f, a-62)	8.40 μmol/L	14.88 μmol/L	*p* < 0.001	S	CCEI	Inclusions: undergoing ocular surgery Exclusions: N/A
[112]	PEXG	Budapest, Hungary (N/A)	18 controls (5 m + 13 f, a-65) 30 in test group (9 m + 21 f, a-69)	9.14 μmol/L	11.95 μmol/L	*p* = 0.023	P	FPIA	Inclusions: cataract surgery, argon laser trabeculoplasty, trabeculectomy Exclusions: N/A
[98]	PEXG	Nuremberg, Erlangen, Germany (N/A)	31 controls (13 m + 18 f, a-72) 29 in test group (13 m + 17 f, a-73)	11.82 μmol/L	15.53 μmol/L	*p* = 0.012	P	ELISA	Inclusions: N/A Exclusions: N/A
[98]	PEXG	Nuremberg, Erlangen, Germany (N/A)	31 controls (13 m + 18 f, a-72) 29 in test group (13 m + 17 f, a-73)	1.26 μmol/L	2.51 μmol/L	*p* < 0.0001	AH	ELISA	Inclusions: N/A Exclusions: N/A
[113]	PEXG	New York, United States (Caucasian, black)	24 controls (10 m + 10 f, a-70) 25 in test group (9 m + 16 f, a-71)	8.3 μmol/L	10.1 μmol/L	*p* = 0.009	P	FPIA	Inclusions: N/A Exclusions: N/A

Note: AH—aqueous humor; ELISA—enzyme-linked immunosorbent assay; P—plasma; FPIA—fluorescence polarization immunoassay; S—serum; CCEI—competitive chemiluminescent enzyme immunoassay; TF—tear fluid; m—male; f—female; a—average of ages; AU—AU 500 automatic biochemical analyzer; N/A—not available from the original paper. ^*1^—controls are subjects without POAG.

**Table 3 ijms-24-10790-t003:** The association of homocysteine levels with NTG.

Studies	Glaucoma Type	Location (Race)	Controls *^1^ (Gender, Age (a)) Test Group (Gender, Age (a))	Hcy Level in Controls	Hcy Level in Test Group	Significance (*p* Value)	Sample Types	Lab Technique	Surgical Eligibility
[106]	NTG	Alicante, Spain (N/A)	75 controls (17 m + 58 f, a-44) 15 in test group (3 m + 12 f, a-45)	5.9 μmol/L	6.4 μmol/L	*p* = 0.002	P	CCEI	Inclusions: N/A Exclusions: N/A
[110]	PEXG + NTG *^2^	Middle Eastern (N/A)	35 controls (18 m + 17 f, a-67) 18 in test group (10 m + 8 f, a-68)	9.9 μmol/L	19.8 μmol/L	*p* < 0.001	P	HPLC	Inclusions: N/A Exclusions: N/A
[108]	NTG	Sydney, Australia (White)	42 controls (16 m + 26 f, a-70) 34 in test group (9 m + 25 f, a-73)	9.82 μmol/L	11.74 μmol/L	*p* < 0.05	P	FPIA	Inclusions: laser trabeculoplasty, glaucoma filtration surgery Exclusions: N/A
[101]	NTG	Middle Eastern Turkish (N/A)	19 controls (5 m + 14 f, a-57) 18 in test group (6 m + 12 f, a-58)	8.40 μmol/L	10.39 μmol/L	*p* > 0.05	S	CCEI	Inclusions: undergoing ocular surgery Exclusions: N/A
[113]	NTG	European American (white, Hispanic, Asian)	24 controls (10 m + 10 f, a-70) 25 in test group (11 m + 11 f, a-70)	8.3 μmol/L	9.1 μmol/L	*p* = 0.2	P	FPIA	Inclusions: undergoing ocular surgery Exclusions: N/A
[114]	NTG	Erlangen, Germany (Caucasian)	42 controls (15 m + 27 f, a-63) 42 in test group (15 m + 27 f, a-66)	11.29 μmol/L	10.95 μmol/L	*p* = 0.639	P	FPIA	Inclusions: N/A Exclusions: N/A

Note: AH—aqueous humor; ELISA—enzyme-linked immunosorbent assay; P—plasma; FPIA–fluorescence polarization immunoassay; S—serum; CCEI—competitive chemiluminescent enzyme immunoassay; TF—tear fluid; m—male; f—female; a—average of ages; AU—AU 500 automatic biochemical analyzer; N/A—not available from the original paper. ^*1^—controls are subjects without POAG. ^*2^—NTG was not assessed independently but rather in patients with concomitant PEXG.

**Table 4 ijms-24-10790-t004:** The association of homocysteine levels with neovascular glaucoma.

Studies	Glaucoma Type	Location (Race)	Controls *^1^ (Gender, Age (a)) Test Group (Gender, Age (a))	Hcy level in Controls	Hcy Level in Test Group	Significance (*p* Value)	Sample Types	Lab Technique	Surgical Eligibility
[100]	Neovascular Glaucoma	Antalya, Turkey (N/A)	30 controls (N/A-mf, a-55 +) 20 in test group (N/A-mf, a-55 +)	10.55 μmol/L	14.99 μmol/L	*p* < 0.0001	P	FPIA	Inclusions: N/A Exclusions: N/A

Note: AH—aqueous humor; ELISA—enzyme-linked immunosorbent assay; P—plasma; FPIA—fluorescence polarization immunoassay; S—serum; CCEI—competitive chemiluminescent enzyme immunoassay; TF—tear fluid; m—male; f—female; a—average of ages; AU—AU 500 automatic biochemical analyzer; N/A—not available from the original paper. ^*1^—controls are subjects without POAG.

## Data Availability

Not applicable.

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
