# Peer review of "Homocysteine and Glaucoma"

_ijms, 2023, doi:10.3390/ijms241310790_

Round 1
Reviewer 1 Report
I believe it is useful to include more details on the causal relationship between hyperhomocysteinemia and glaucoma in the "Conclusions".
Additional suggestions could also be made for the prevention and therapy of glaucoma by regulation of homocysteine levels and for new lines of research.
Author Response
- More details about the relationship between hyperhomocysteinemia and glaucoma have been included. We mentioned causes that stem from glaucoma-relevant genetics and environmental factors when discussing the development of glaucoma.
- Additional suggestions for glaucoma treatment have been added to the conclusion. This includes changes in diet habits, medication adherence, and annual ophthalmic exams for glaucoma prevention. We also mentioned how seeking standard care could provide novel approaches for clinical treatments.
Reviewer 2 Report
This manuscript is a comprehensive review article summarizing the current efforts on elucidating the correlation between elevated homocysteine level and glaucoma pathogenesis. The authors provide wide-ranging references including human and animal data and controversial results for discussion, which is appreciated. In general, the manuscript is well-written and easy to follow. Major corrections for figures and table as well as minor comments are listed below.
1. All figures were not original. Although the authors cited reference for the source, it is not sure if they received the permission to directly use these figures. A proper acknowledgment should be provided in the manuscript to avoid copyright issues; an agreement statement would be better. Otherwise, the authors should consider creating their own figures (see below comments).
2. Figures 1-3 are helpful to understand Hcy metabolism. However, too many acronyms need to define in the legend. The authors should consider either providing more detailed information in the legend or making simple versions of these pathways.
3. The font size in current Figures 4-6 is too small to read, and a much more detailed figure legends are needed to help readers appreciate the concept. The reviewer would strongly suggest the authors to re-make and simplify these figures to exclude unnecessary details and avoid the confusion. Perhaps the authors can consider combining Figures 4 and 6 and just point out key components in eye structure with the bottom right of figure 6 (retina cell layers).
4. In most cases, the authors use abbreviation for specific type of glaucoma, while in some cases (e.g. line 257) the full name was used. Please keep it consistent.
5. Many sections in this manuscript specifically called out POAG. Such as lines 293-297 and section 9. Perhaps the authors can add one or few sentences in Section 6, when introducing various types of glaucoma, to provide some perspectives on why focusing on POAG (e.g. most common form of glaucoma, etc).
6. Suggested re-organizing Table 1.
a. Consider grouping the studies by "glaucoma type" - current order is pretty random and hard to follow. It is suggested have all POAG in one section, followed by NTG or PEXG, etc. Of course, there will be duplicate references since some studies assessed more than one types of glaucoma.
b. Consider adding other information - human or animal (rodent?) data; age; gender; unilateral or bilateral observation; with or without surgery, etc.
c. The reviewer did not check over all the references in Table 1. However, by spot checking, the sample sizes for cases/controls are reversed in several studies, such as studies 98, 99, 102. Please ensure the numbers are correct for all studies in table 1.
d. The population column might be misleading as these studies were conducted in certain location, not necessarily indicating the races. For example, study 102 was conducted in Australia and most participants were white; however, few Asian and others were included as well. Thus, the authors stated "Australian" may be misleading. It is suggested replacing this column with "location" and if possible, adding a new column for the race. Noted it is possible that not all studies include races in the metadata and that should be shown as N/A.
e. The definition of "control" can vary in different studies. For example, study 98 indicated that control groups are patients with cataracts. Since the authors also stated in this manuscript that surgery can be a risk factor for glaucoma progression, it should be clarified in the table whether the control group in individual study means healthy, no ocular diseases, without glaucoma, without glaucoma but experiencing ocular surgery, etc.
f. In study 109, the Table shows 55 POAG and 55 controls were examined, and this was also stated in line 318. The original report stated that 39 control (no ocular disease, healthy, cataract extraction), 16 non-POAG and 55 POAG were included. Please clarify.
g. There are few lines in the table for studies 99, 101. Assuming these are formatting errors.
h. In study 99, p value for NTG group shows "p*1". Please define "*1". Same comment for Glaucoma type in study 100 showing "NTG*2"
i. On the top row of Table 1, there are extra spaces between words, such as Sample Size, Hcy Level, and Used Lab. In study 94, extra spaces were also present in between 14.99. Please remove these extra spaces.
7. The authors might consider commenting on how various sample types (serum/plasma vs aqueous humor) or lab assays (HPLC vs ELISA) could potentially impact the measurement and data interpretation in Table 1.
8. This might be nitpicking. Because this review aims to answer the question whether HHcy is a risk factor for glaucoma, with controversial data to date, the reviewer recommended the term "HHcy-associated POAG" throughout the text. The term "HHcy-induced POAG" is kind of self-stated a positive correlation that Hcy is a causative agent.
9. Line 379: This study (#131) involves various mouse models including Nrf2-null mice and the quantitative RGC viability assay was performed in vitro using RGC and Muller cell co-cultures isolated from wild-type and Nrf2-null mice. Please specify which model that the authors were referring to and be clear about the contribution of NRF2.
10. Line 389: What data is “this data”?
11. Editorial suggestions:
a. Line 51: suggesting "inconclusive" or "controversial" to replace “unknown”.
b. Line 110: suggested "exhibits" to replace “has”
